# Propensity of *Tagetes erecta* L., a Medicinal Plant Commonly Used in Diabetes Management, to Accumulate Perfluoroalkyl Substances

**DOI:** 10.3390/toxics7010018

**Published:** 2019-03-25

**Authors:** John Baptist Nzukizi Mudumbi, Adegbenro Peter Daso, Okechukwu Jonathan Okonkwo, Seteno Karabo Obed Ntwampe, Tandi E. Matsha, Lukhanyo Mekuto, Elie Fereche Itoba-Tombo, Adewole T. Adetunji, Linda L. Sibali

**Affiliations:** 1Bioresource Engineering Research Group (BioERG), Department of Biotechnology and Consumer Sciences, Cape Peninsula University of Technology, PO Box 652, Cape Town 8000, Western Cape, South Africa; NtwampeS@cput.ac.za (S.K.O.N.); elie.tombo@gmail.com (E.F.I.-T.); 2Department of Environmental, Water and Earth Sciences, Faculty of Science, Tshwane University of Technology, Pretoria 0083, South Africa; adegbenrop@gmail.com (A.P.D.); OkonkwoOJ@tut.ac.za (O.J.O.); 3Department of Bio-Medical Sciences, Faculty of Health and Wellness Science, Cape Peninsula University of Technology, PO Box 1906, Bellville 7535, Western Cape, South Africa; MATSHAT@cput.ac.za; 4Department of Chemical Engineering, University of Johannesburg, PO Box 17011, Johannesburg 2028, Gauteng, South Africa; lukhayo.mekuto@gmail.com; 5Department of Agriculture, Cape Peninsula University of Technology, Wellington Campus, Wellington 7655, Western Cape, South Africa; adetunjiadewole@gmail.com; 6Research Management Unit, Faculty of Applied Sciences, Cape Peninsula University of Technology, PO Box 652, Cape Town 8000, Western Cape, South Africa; SibaliL@cput.ac.za

**Keywords:** medicinal plants, perfluoroalkyl substances (PFASs), perfluorooctanoic acid (PFOA), perfluorooctane sulfonate (PFOS), perfluorobutane sulfonate (PFBS), *Tagetes erecta* L.

## Abstract

It has been extensively demonstrated that plants accumulate organic substances emanating from various sources, including soil and water. This fact suggests the potentiality of contamination of certain vital bioresources, such as medicinal plants, by persistent contaminants, such as perfluorooctanoic acid (PFOA), perfluorooctane sulfonate (PFOS), and perfluorobutane sulfonate (PFBS). Hence, in this study, the propensity of *Tagetes erecta* L. (a commonly used medicinal plant) to accumulate PFOA, PFOS, and PFBS was determined using liquid chromatography/tandem mass spectrometry (LC–MS/MS-8030). From the results, PFOA, PFOS, and PFBS were detected in all the plant samples and concentration levels were found to be 94.83 ng/g, 5.03 ng/g, and 1.44 ng/g, respectively, with bioconcentration factor (BCF) ranges of 1.30 to 2.57, 13.67 to 72.33, and 0.16 to 0.31, respectively. Little evidence exists on the bioaccumulative susceptibility of medicinal plants to these persistent organic pollutants (POPs). These results suggest that these medicinal plants (in particular, *Tagetes erecta* L., used for the management of diabetes) are also potential conduits of PFOA, PFOS, and PFBS into humans.

## 1. Introduction

Evidence exists which indicates that plants were used for medical purposes long before the industrial epoch. Ancient Egyptian papyrus manuscripts have also reported and suggested the extensive use of medicinal plants. Currently, the World Health Organization (WHO) has estimated that 80% of the global population rely on medicinal plants for aspects of their first-hand health care requirements [1]. African marigold (*Tagetes erecta* L.) is a member of the Asteraceae plant family. Evidence has indicated that *Tagetes erecta* L. is well-known as an important commercial plant utilized mostly for decorative purposes [2,3,4]. Recently, the plant has been renowned for its industrial and medicinal usage [5,6,7]; a number of studies have suggested that *Tagetes erecta* L. has the potential to treat ailments such as diabetes mellitus (DM) [8,9,10,11,12]. In South Africa, use of the leaves of *Tagetes erecta* L. in the treatment of DM has been reported [13].

Nevertheless, these phyto-bioresources are believed to be susceptible to environmental effects, including negative externalities such as contamination by toxic substances, especially persistent organic pollutants (POPs). This assertion is based largely on evidence indicating that plants are capable of taking up and accumulating nutrients and a variety of other chemicals to which they are, either directly or indirectly, exposed. Thus, compelling evidence has demonstrated that plants accumulate and metabolize environmental contaminants, ultimately suggesting that plants are reservoirs for chemical substances [14,15]. Some scientists have reported the prevalence of toxic substances and/or heavy metals in plants [16,17,18,19,20,21,22,23,24]. Moreover, various medicinal plants have previously been reported to be threatened by exposure to chemical substances, including heavy metals. For instance, research results have recently suggested that medicinal plants’ exposure to chemical substance results in chemo-stress, which influences the antioxidant status of the plant and culminates in damage to its DNA [25].

Previously, heavy metals, including barium (Ba), chromium (Cr), cadmium (Cd), iron (Fe), strontium (Sr), lead (Pb), and zinc (Zn) have been reported in medicinal plants [15,26]. Furthermore, a study by Tian et al. [27] determined that plant leaves are effective in taking up PFASs from the atmosphere, with previous studies by Blaine et al. [28] reporting the bioaccumulation of various perfluoroalkyl acids (PFAAs) in edible crops, including lettuce (*Lactuca sativa)* and strawberry (*Fragaria ananassa*), suggesting these crops are a potential route of exposure for humans. In most instances, it is contaminated river water and fertilizer, as well as aero-deposition, that results in the contamination of these plants [29,30]. Nevertheless, due to limited available evidence on the contamination of medicinal plants by PFASs [15], the possibility that these plants are a pathway through which humans are likely to be exposed to PFASs is still to be established. It is worth noting that available evidence has reported wide concerns about these substances, and their health safety remains unclear [31,32,33,34]. Nevertheless, health advisory standards have been proven [34], and can be used as a benchmark for the establishment of a better safety level for toxicity of these substances for humans. Therefore, the aim of this study was to determine the propensity of *Tagetes erecta* L., a common medicinal plant used by diabetic patients in sub-Saharan Africa, to accumulate PFOA, PFOS, and PFBS.

## 2. Materials and Methods

### 2.1. Chemicals and Reagents

A specific perfluorocarboxylic acid (PFCA) standard (i.e., perfluorooctanoic acid (PFOA)), and singular linear perfluoroalkyl sulfonic acids (PFSAs) such as perfluorobutane sulfonate (PFBS) and perfluorooctane sulfonate (PFOS), were obtained from the laboratory facility of the Department of Environmental, Water and Earth Sciences, Tshwane University of Technology (TUT), South Africa; these were purchased in methanol at 50 µg/mL from Wellington Laboratories (Ontario, Canada). A solution of surrogate mixture of stable isotopically-labelled PFAS standard containing perfluoro-n-[1,2,3,4-^13^C_4_] octanoic acid (MPFOA), perfluoro-n-[1,2,3,4, 5-^13^C_5_] nonanoic acid (MPFNA), and perfluoro-n-[1,2-^13^C_2_] undecanoic acid (MPFUnDA) was also obtained from TUT, and purchased in methanol at 50 µg/mL from Wellington Laboratories (Ontario, Canada). Acetic acid, polypropylene (PP) membrane filters (0.22 µm, Cameo syringe filters) and syringes (Becton Dickinson), LC–MS grade water, acetonitrile, methanol, and ammonium acetate, as well as Supelco-Select HLB SPE cartridges (500 mg), were purchased from Sigma-Aldrich (Aston Manor, South Africa). T Milli-Q water was used throughout the study.

### 2.2. Sample Collection: Tagetes erecta L. and River Water

Samples of plant leaves (*n* = 8) were harvested from main plants (i.e., *Tagetes erecta* L.) separated in cultivation pots. River water samples (*n* = 20) from the Salt River, Western Cape, South Africa, were used to irrigate the plants. The river water samples were randomly taken during summer months (i.e., dry season—March) and winter months (i.e., wet season—August), with the bulk of the river water being used to irrigate the plants without pre-treatment at a frequency of 120 mL every two to three days for pots containing 0.5 L of loamy soil.

### 2.3. Sample Pre-Treatment and Solid Phase Extraction

#### 2.3.1. Plant Samples

Samples were pre-treated using protocols previously used by Tian et al. [27] and Mudumbi et al. [28], with minor changes. Thus, plant leaf samples (*n* = 8) were harvested using a laboratory scalpel and oven-dried for 24 h at approximately 60 °C, and subsequently milled into a powder form. Thereafter, 2 g from each of the samples was transferred to a clean 15 mL PP centrifuge tube. The tubes were subsequently spiked with a 50 µL surrogate mixture of stable isotopically-labelled PFASs standard (i.e., MPFOA, MPFNA, and MPFUnDA), and the mixture was allowed to equilibrate for about 1 h at ambient temperature (21–26 °C). Subsequently, 15 mL of 0.01 M NaOH/MeOH was added and the mixture was then homogenized by vigorous vortexing (2 min), at ambient temperature. Subsequently, the PP tubes were centrifuged at 3000 rpm for 4 min and the supernatants were emptied into new PP tubes (15 mL) pre-rinsed with analytical LC–MS grade methanol. The cycle was repeated twice, and the supernatants from both cycles were filtered using polypropylene 0.22 µm Cameo syringe filters (Sigma-Aldrich, Darmstadt, Germany). Thereafter, a total volume of 15 mL was recorded, which was used for solid phase extraction (SPE).

#### 2.3.2. River Water Samples

River water was randomly collected in PP containers of 25 L capacity, from a local Western Cape river (i.e., Salt River) previously known to be contaminated with PFASs [29], and the PFASs analyses were carried out based on the same source protocols, with negligible changes. Hence, from this water, a total of twenty samples (*n* = 20) was randomly taken from the river water to irrigate the plants. The samples contained suspended particulate matter (SPM), which was removed by means of filtration; PP membrane filters (0.22 µm, Cameo syringe filters, Sigma Aldrich, Darmstadt, Germany) were used. Subsequently, the filtered river water samples were spiked with 50 µL of a surrogate mixture of stable isotopically-labelled PFASs standard (i.e., MPFOA, MPFNA, and MPFUnDA), and vortexed (2 min) prior to SPE, without pH adjustment or dilution.

#### 2.3.3. Solid Phase Extraction

Supelco-Select HLB SPE cartridges (500 mg solid phase, 12 mL tubes) were used for SPE using procedures as suggested in previous studies, including Mudumbi et al. [28,29,30], with minor modifications. Hence, the cartridges were preconditioned with 5 mL of analytical LC–MS grade methanol and then 5 mL of Milli-Q water at a flow rate of 1 drop per two seconds. After loading the samples (i.e., a volume of 15 and 20 mL of plant and water extracts, respectively) at a flow rate of one drop per two seconds, Supelco-Select HLB SPE cartridges were washed with 5 mL of 40% (*v*/*v*) analytical LC–MS grade methanol in Milli-Q water, as reported by Mudumbi et al. [28,29]. Successively, PP collection tubes were added to the SPE apparatus, and PFASs were eluted from Supelco-Select HLB SPE cartridges into the PP collection tubes, using 10 mL of analytical LC–MS grade methanol. It was extremely pertinent to use PP collection tubes in order to minimize background cross-contamination of the eluents. The tubes were thereafter dried under nitrogen gas, and reconstituted with 0.5 mL of 50 ng/mL M2PFOA internal standards (ISTD) prepared in 10% acetonitrile. Figure 1 outlines the scheme of the overall process used. The final aliquots (500 µL) of the supernatants were transferred into PP autosampler vials before analysis using LC–MS/MS.

### 2.4. LCMS-8030 Analysis

#### 2.4.1. LCMS-8030 Configuration for PFOA, PFOS and PFBS Quantification

The analysis of PFASs (i.e., PFOA, PFOS, and PFBS) in plant and river water samples was conducted using a liquid chromatograph (LC) coupled with triple quadrupole linear ion trap tandem mass spectrometer (Shimadzu LCMS-8030, Canby, OR, USA) equipped with an electrospray ionization (ESI) source, which was in a negative ion mode. The targeted PFASs were quantified using multiple reaction monitoring (MRM) mode of analysis. The chromatographic separation of analytes was achieved with a Luna^®^ Omega Polar C18 column (2.1 × 100 mm, 3.0 µm, Phenomenex, Aschaffenburg, Germany). The column temperature was set at 40 °C. A gradient elution program was applied and was made of 20 mM ammonium acetate (solvent A) and 100% MeOH (solvent B), at a flow rate of 0.3 mL/min and an injection volume of 10 µL used for individual samples. The linear gradient elution program started at 20% B and increased to 80% B after 5 min, then increased to 95% B for 15 min; it was kept to 100% B for 17–27 min, before being 20% B for 30–40 min. The total run time for each injection was 40 min. Argon gas was used as the collision gas. The LC system was a LCMS-8030 Shimadzu system with a DGU-20A_3R_ degassing unit, coupled with an LC-20AD liquid chromatograph, a CTO-20AC column oven, a SIL-20AC autosampler and a NM32LA nitrogen gas generator.

#### 2.4.2. Validation of Method

To ensure method precision, procedural blanks were prepared during the analysis and were analyzed at an interval of ten samples. This was to assess whether contamination occurred during sample extraction. Hence, solvent blanks comprising MeOH (195 µL) and ISTD (5 µL) were prepared for analyses after every twenty processed samples to monitor for background contamination. To assure the accuracy and precision of each run, duplicate injections and recalibration using appropriate standards were conducted for each run after processing twenty samples. In cases whereby the target analytes were detected in the procedural blanks, their peak areas’ average values were subtracted from the peak areas of the target analyte of the actual sample before the final concentrations were calculated. The level of detection (LOD) was defined as the peak signal of a target analyte that needed to yield a signal-to-noise (S/N) ratio of 3:1 and ranged from 0.003 to 0.03 ng/L for all the three investigated PFASs. The limit of quantification (LOQ), was defined as the standard deviation (SD) of the blanks and was determined to be 0.03 ng/L for PFOA and PFOS, and 0.07 ng/L for PFBS. Additionally, 50 µL of native surrogates were used for matrix spike recovery testing. Hence, recoveries of native standard surrogates spiked in the plant and water matrix were 98, 96, and 93% for PFOA, PFOS, and PFBS, respectively. Furthermore, Equations (1) and (2) were used to obtain the relative response factors and final concentrations of the targeted PFASs, respectively.
(1)$$RRF=\frac{A_{\mathrm{NAT}}}{A_{\mathrm{IS}}}\times \frac{C_{\mathrm{IS}}}{C_{\mathrm{NAT}}}$$
where:
*RRF* is the relative response factor;*A*_NAT_ is peak the area of the native compound;*A*_IS_ is the peak area of the internal standard in the standard;*C*_NAT_ is the concentration of the native standard;*C*_IS_ is the internal standard concentration.
(2)$$FC=\frac{A_{\mathrm{NAT}}}{A_{\mathrm{IS}}}\times \frac{1}{RRF}\times \frac{V_{\mathrm{IS}}}{V_{S}}$$
where:
*FC* is the final concentration;*A*_NAT_ is the peak area of the target analyte;*A*_IS_ represents the peak area of the internal standard used for that particular analyte;*RRF* is the calculated relative response factor of the specific analyte;*V*_IS_ is the volume of the internal standard added in the sample prior to extraction (mL);*V*_S_ is the volume of the sample (mL).

## 3. Results

### 3.1. LCMS Calibration Curves for the Detection and Quantification of PFOA, PFOS and PFBS

A procedural blank matrix free of the 3 PFASs was prepared and used in preparation for post-spiked calibrants, and thus the calibration curves were constructed based on a 10-point curve at concentrations of 1, 2, 5, 10, 20, 25, 50, 75, 100, and 125 ng/L. The regression coefficients (*R*^2^) of calibration curves for all the target analytes have revealed good linearity (*R*^2^ > 0.99), as can be seen in Figure 2 which displays the calibration curves of PFOA, PFOS, and PFBS.

### 3.2. LCMS Chromatographs for PFOA, PFOS, and PFBS

The MRM optimization of three PFASs (i.e., PFOA, PFOS, and PFBS) and one ISTD (i.e., M-PFOA) was carried out, with two MRM transitions being utilized for each PFAS. Thus, one was used as an ion quantifier and the other for confirmation. Table 1, as well as Appendix A, shows the mass transitions used for the identification and quantification of each targeted compound, as well as the ISTD, and their retention times (RT).

### 3.3. Results of Previously Known Contaminated River Water

Although evidence of PFASs in the South African environment remains limited, a previous study has reported concentrations of PFOA and PFOS in a Western Cape river (i.e., Salt River) of 0.7 to 390 ng/L and <LOD to 50 ng/L, respectively [29]. Of the three rivers which were studied for their PFASs predisposition, the Salt River recorded the highest PFOA concentration. The Salt River also had the second-highest PFOS concentration, although PFBS was not investigated. In this current study, the water which was collected from the Salt River was for the purpose of irrigation of the plants that were studied. Therefore, it was pertinent to first assess the concentration levels of PFASs in the collected water, prior to using the water for irrigation purposes, and to ensure the accuracy of the results. Therefore, three PFASs (i.e., PFOA, PFOS, and PFBS) were quantified in twenty samples (*n* = 20). Two sampling regimes were implemented with river water: Regime A (*n* = 10) samples were taken after heavy rain, and constituted winter/wet season conditions, while Regime B (*n* = 10) samples were taken during the summer/dry season, for which rainfall was absent for the previous five months. The results obtained in this regard are summarized in Table 2, and it can clearly be seen from these that the investigated substances have been detected in some samples. From the investigated plant samples, the concentration of the substances varied markedly between individual samples, as well as the river water regimes. The PFAS concentrations in samples were in the following decreasing order: PFOA > PFBS > PFOS. From the investigated samples, Regime A registered all the highest concentrations in terms of the analyzed substances, while Regime B recorded the lowest. On the other hand, Figure 3 demonstrates how each river water sample has contributed to the overall concentrations of each investigated substance.

### 3.4. PFOA, PFOS and PFBS Accumulation in a Commonly-Used Medicinal Plant

There are various reports which have indicated the prevalence of PFASs (i.e., PFOA, PFOS, and PFBS) in several environmental matrices, including plants. For instance, Mudumbi et al. [28] reported the susceptibility of riparian plants to PFOA accumulation in South Africa, Western Cape Province (WCP), while Krippner et al. [35,36] indicated higher uptake of PFASs, including PFBS, into plant leaves. Recently, Kurwadkar et al. [37], as well as Zhao and Zhu [38], addressed the uptake of PFASs in plants. Similarly, studies by Sznajder-Katarzyńska et al. [39] and Zhao et al. [40] have reported on the vulnerability of edible plants to accumulation of PFASs. Nevertheless, there is little evidence on the vulnerability of medicinal plants to PFASs accumulation [15], as most studies have focused on the therapeutic side of these plants and not on their susceptibility to emerging POPs, such as PFASs, which are a potential risk to human health. For this reason, PFASs (i.e., PFOA, PFOS, and PFBS) were investigated in *Tagetes erecta* L., and traces of the three PFASs were detected in all the plant samples. The concentrations of these POPs among all the investigated plant samples were in the following decreasing order: PFOA > PFOS > PFBS. Contaminated samples recorded the highest amount of PFOA and PFOS. The summary of these results is depicted in Table 3, and Figure 4 shows the contribution of each sample to the concentration levels of PFASs that were quantified in the plant under investigation.

## 4. Discussion

### 4.1. New Evidence on the Contamination of Salt River by PFASs

Concentrations of PFOA, PFOS, and PFBS were observed in all the samples, with PFBS being the most dominant PFAS, followed by PFOA. However, concentration levels for PFOS were mostly not detected (ND) for individual samples. The results are summarized in Table 2. From the results, it can be seen that the concentrations of PFOA, PFBS, and PFOS were <LOD to 107.82 ng/L; 1.24 to 20.75 ng/L; and ND to 0.12 ng/L, respectively. Overall, Regime A samples had the highest concentrations of PFASs with sample RW5 having 107.82 ng/mL for PFOA, RW2 20.75 ng/L for PFBS, and RW3 0.12 ng/L for PFOS. However, the second sample (i.e., RW11) had the highest PFOS concentration (0.10 ng/L) observed among the Regime B samples. Figure 3 shows PFAS concentration variations in samples from the two regimes, A and B, (i.e., for samples taken in two different seasons).

Furthermore, from Table 2, it can be seen that two of the three assessed PFASs (PFOA and PFBS) showed a significant increase during Regime A, which was putatively regarded as a result of the rain which might have contributed to run-off of PFAs into the river. This trend substantiates the fact that runoff has been suggested as being a contributing factor to higher concentrations of PFASs in water streams [29,41]. Overall, PFBS was prevalent in most samples, although PFOA was observed to have had the highest concentrations in a few samples, with the PFOA concentrations of most samples being below the LOQ (that is, 0.03 ng/L). Similarly, PFOS concentration levels remained below the LOQ in some samples (*n* = 7), with only one sample (RW5) being below the LOD (that is, 003 to 0.03 ng/L). Additionally, PFOS was the only PFAS that was not detected in certain individual samples, including sample RW2. PFBS was found to be prevalent in both sampling regimes (A and B), while LOQ for PFOA and PFOS were evenly distributed, in particular for Regime B. As both PFOA and PFOS are classified as long-chained PFASs, while PFBS is identified as a short-chain compound [42], it was previously suggested that PFOA and PFOS prevalence in the Western Cape rivers might be attributed to a highly active agricultural sector [29]. These two PFASs have been the most studied and have predominantly been found in various environmental matrices, both worldwide and in South Africa [14]. Recent reports have now indicated that PFBS, previously thought to be harmless, fits the category of POPs [14]. In addition, recent reports have now indicated that PFBS, previously known as a harmless PFAS, fits the category of POPs [14,43,44,45], and it has been found to be the most dominant PFAS in this study—a pattern previously reported by Heydebreck et al. [46] and Pan et al. [47]. This ultimately suggests the use of PFBS in the Western Cape, South Africa, and thus, there is cause for concern with regard to the prevalence of this short-chain PFAS in the South African environmental ecosystem, especially in river water. Accordingly, further studies are required to determine other short-chain PFASs prevalent in the South African environment, and their possible source(s). Nevertheless, Cai et al. [41] and Zhu et al. [48] have reported that the abundance of short-chain PFASs signifies the predominance of the use of perfluorocarboxyl compounds in a study area. Evidence of the prevalence of short-chain PFASs in humans is also limited (if not non-existent) in the Western Cape, and particularly in South Africa.

Furthermore, we compared the concentration levels of the three PFASs investigated in the Salt River with those found in other rivers worldwide (see Table 4). As far as the Salt River is concerned, it was found that concentrations of PFOA and PFOS were much lower than they were in previous studies conducted in 2014, and remained the lowest among comparative PFASs studied [29]. This decrease can be attributed to the fact that during the sampling year for this study (2017), the Western Cape Province experienced a severe drought, which led to minimal and/or limited runoffs into the river under investigation. It was further suggested that there has been a decrease in the use of the said substances and/or products containing them in the region. This argument still has to be confirmed by further investigations. Nevertheless, the concentration levels of both PFOA and PFBS, in the current study, were found to be much higher than in other rivers globally, but PFOS concentration remained generally much lower, or undetected. These results are similar to those of the Rhine River (see Table 4), and the PFBS concentration determined in this study was also similar to that of the Rhine river [46].

### 4.2. Traces of PFASs in the Investigated Medicinal Plant

In this study, the propensity of the African marigold (*Tagetes erecta* L.) to accumulate PFOA, PFOS, and PFBS was investigated. *Tagetes erecta* L. is a medicinal plant commonly used for DM therapy [8,9,10,11,12,13]. Since the study was conducted using a set of plants, we used contaminant-free plant sets as a reference. The soil in which the plants were grown was not assessed for PFASs as they were grown in pristine soil, with the source of the PFAS being the river water.

Subsequently, PFOA, PFOS, and PFBS, as found in the river water, were observed in all the plant samples (*n*= 8) with PFOA being the most highly accumulated PFAS by *Tagetes erecta* L., followed by PFOS, and then PFBS, with concentrations of up to 94.83 ng/g, 5.03 ng/g, and 1.44 ng/g, respectively. Table 3 displays the overview of these concentrations. In addition, these concentrations were attributed to the highest concentration of both PFOA and PFBS in the river water, hence their prevalence in higher concentration in the plant samples. The accumulation was hypothesised to be facilitated by mass flow translocation, a process through which chemical constituents in water are taken up by the plants [55,56,57] via the root system of the plant [14,56,57]. Hence, it can be suggested that the higher the concentration of PFASs in the water, the higher the likelihood of these pollutants to accumulate in plant compartments, including leaves. These results are an indication that medicinal plants are at risk of being contaminated by pollutants, including PFASs, and ultimately, constitute a potential pathway through which these substances might be ingested by humans who rely on them for therapeutic purposes. Hence, Appendix A depicts a list of select medicinal plants that are used to treat T2DM in South Africa, which are at risk of being exposed to the prevalence of PFASs, as river water is predominantly used in underprivileged communities which rely heavily on phytomedicines for the management of diseases.

Furthermore, the results obtained in the current study partially concur with the results previously found by Mudumbi et al. [28], Yoo et al. [58], Marchand et al. [59], and Stahl et al. [60], which reported that various plants had the potential to accumulate PFASs, PFOA in particular. However, a slight decrease in the uptake of PFOA was observed in the present results compared to that reported by Mudumbi et al. [28]. As previously suggested, the contribution of the root system of the studied plant, that is *Tagetes erecta* L., to the uptake of PFASs was not analysed, a factor which Mudumbi et al. [28] suggested to play a pivotal role in the manner in which a given plant uptakes pollutants, including PFASs.

### 4.3. Tagetes erecta L. Sorption Aptitude by Means of Bioconcentration Factor (BCF)

Bioconcentration factor (BCF), according to available evidence, is seen as the capability of a plant to uptake a specific chemical substance with relation to its concentration in the soil [61,62]. Hence, the BCF, in this regard, was calculated as the ratio of the concentrations of the PFASs in the plant samples to those in the river water samples to assess the sorption capacity of *Tagetes erecta* L.:BCF = C_plant samples_/C_water_(3)

Consequently, the BCFs of PFASs for the investigated plant species (i.e., *Tagetes erecta* L.) are shown in Table 3. Hence, for PFOA, the BCF for the different plant samples was 1.30 (CS1), 1.57 (CS2), 2.52 (CS3), 0.86 (S4), 0.92 (S5), 0.99 (S6), 0.76 (S7), and 0.48 (S8); for PFBS it was 0.16 (CS1), 0.31 (CS2), 0.24 (CS3), 0.31 (S4), 0.05 (S5), 0.16 (S6), 0.10 (S7), and 0.11 (S8), while for PFOS, it was 13.67 (CS1), 43 (CS2), 72.33 (CS3), 4 (S4), 119 (S5), 167.67 (S6), 141.33 (S7), and 46.33 (S8). Overall the BCF values for PFOS were higher than those of PFOA and PFBS, a trend which suggests that there was a bioaccumulation potential of this particular PFAS in *Tagetes erecta* L., when compared to the other two PFASs. In this regard, individual plant samples demonstrated an accumulation potential of PFOS. Not only plants were determined to accumulate PFASs in South Africa, another previous study indicated the predominance of PFASs in South African drinking water sources [63], suggesting that even when tap water is used for irrigation, there would be a potential of PFAS accumulation in the plants

Furthermore, PFBS, which is a short-chain PFAS, tends to demonstrate much lower adsorption potential than PFOS and PFOA, which are long-chained PFASs, ultimately suggesting that their bioaccumulation potential in plants might be dependent on their molecular size, as previously suggested by Zhou et al. [64] and Conder et al. [65]. Additionally, it has been indicated that PFBS tend to translocate horizontally and vertically with water diffusion and permeation, making it a much more mobile PFAS than PFOA and PFOS [64]. In addition, the BCF of two (i.e., PFOA and PFBS) of the three investigated PFASs has remained slightly high in the contaminated plant samples. It has been previously suggested that the distribution and accumulation of PFAS in plants are species-dependent [29,66].

### 4.4. Environmental Implications

Subsequently, the benefits of medicinal plants and their hypoglycemic effects in the management of T2DM have overwhelmingly been confirmed by an assortment of studies [15,67,68,69,70]. Nevertheless, evidence on the vulnerability of medicinal plants to pollutant accumulation, including the emerging ones, such as the PFASs, remains limited. This constitutes a cause for concern; according to Mudumbi et al. [15], medicinal plants have played a tremendous role in battle against several diseases, particularly in the sub-Saharan African region, due to the prohibitive cost of orthodox medicine and the low incomes of many communities in the region [71]. This suggests that medicinal plants and/or their derived products are accessible and affordable to these communities [1,2,3,4,5,6,7,8,9,10,11,12,13,14,15]. Hence, Mudumbi et al. [15] suggested that the cultivation, harvest or collection, and storing of medicinal plants and/or their products should be conducted in areas free of any form of contamination, including that of PFASs. The authors further argued that this precautionary measure would ensure enhanced quality, efficacy, and safety of medicinal plants and/or products, and eventually enhanced health for those who rely on these plants as a means of treatment for the ailments they are suffering from, such as T2DM. Moreover, although the future of medicinal plants is promising in the Sub-Saharan region, there is a need for education around conservation, and awareness as to the dangers of using contaminated river water for irrigation purposes [72].

## 5. Conclusions

South Africa is a water-stressed country with uncontrolled contamination of the river water, particularly in certain provinces such as the Western Cape, which experienced a severe drought recently. Subsequently, it has been reported that surface and tap water, as well as riparian plants, in the Western Cape region are contaminated with emerging pollutants, such as PFASs. In the present study, river water was used to irrigate a medicinal plant used to manage DM, *Tagetes erecta* L., as is commonly done in local communities. The PFASs levels in this water were also analysed, as well as the tendency of this plant (i.e., *Tagetes erecta* L.) to uptake these compounds. Consequently, PFOA, PFOS, and PFBS were found in the river water, as well as in the plant under investigation. Individual plant samples demonstrated abundant PFOA concentrations, thus bioaccumulation, and PFBS was observed to be the most predominant in all the river water samples. The BCF suggested that PFBS, a short-chain PFAS, has lower translocation potential into the plant, a trend which allowed this PFAS to remain in the water. In addition, the relatively low accumulation of PFOS in the plant was hypothesized to be dependent on plant species, but future studies still have to be conducted in this regard. Moreover, the prevalence of PFASs in river water used for irrigation, and their subsequent bioaccumulation in medicinal plants, can be considered as a potential pathway through which humans can be exposed to PFASs in communities relying on alternative and unorthodox management of DM. The results from this current study can contribute to the establishment of a database for monitoring the accumulation of PFASs, including PFOA, PFOS, and PFBS, in medicinal plants. There is currently limited information on their susceptibility to PFASs, such PFOA, PFOS, and PFBS, and there is more that still needs to be established.

## Figures and Tables

**Figure 1 toxics-07-00018-f001:**
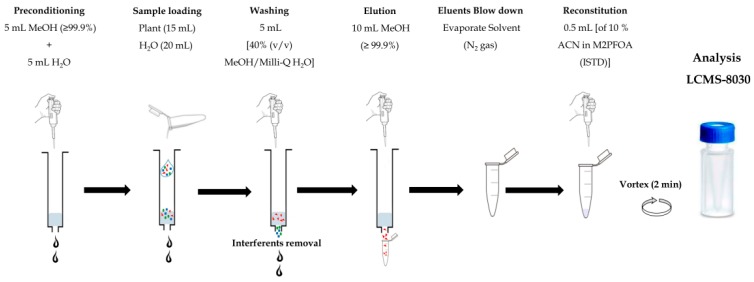
Schema for solid phase extraction (SPE) of water and plant samples.

**Figure 2 toxics-07-00018-f002:**
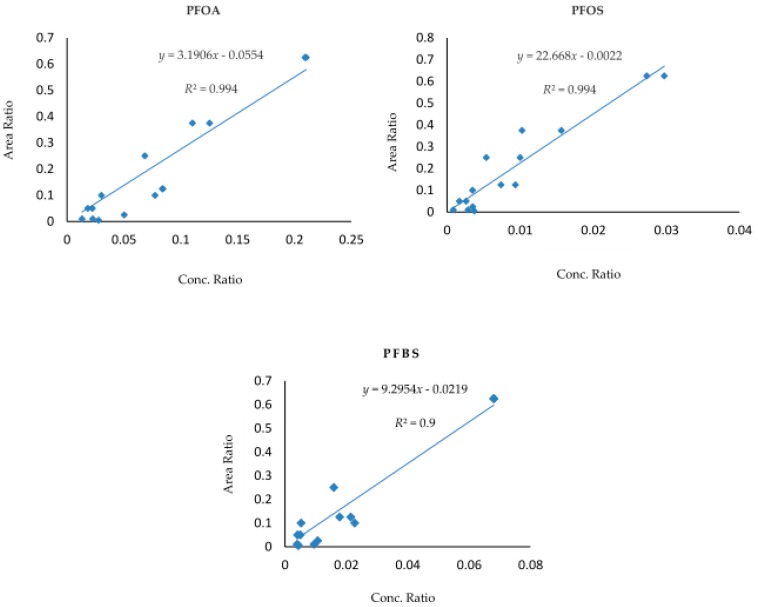
Calibration curves (ng/L) of perfluorooctanoic acid (PFOA), perfluorooctane sulfonate (PFOS), and perfluorobutane sulfonate (PFBS) in procedural blank matrix.

**Figure 3 toxics-07-00018-f003:**
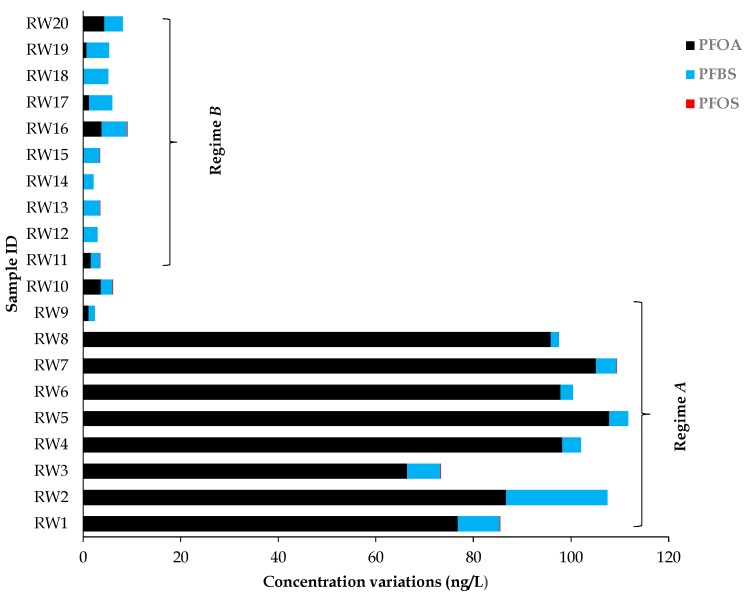
Individual PFAS concentration level variations for each sampling regime.

**Figure 4 toxics-07-00018-f004:**
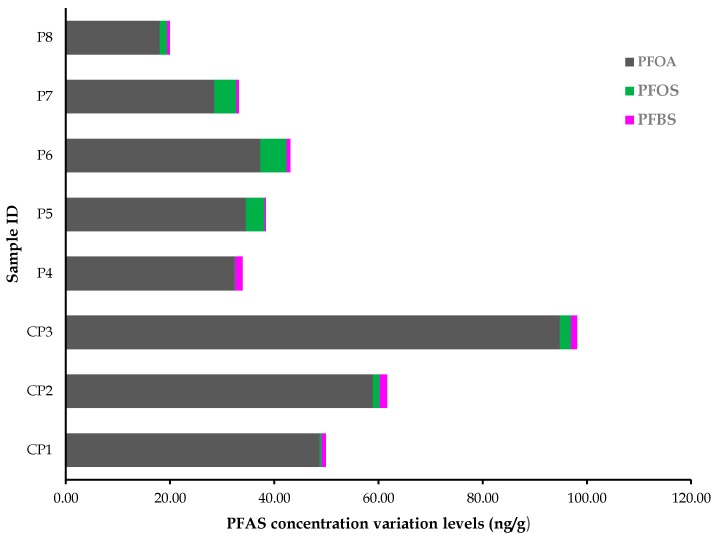
Contribution of each sample to the PFASs concentration levels in *Tagetes erecta* L.

**Table 1 toxics-07-00018-t001:** Names and multiple reaction monitoring (MRM) transitions of three perfluoroalkyl substances (PFASs) and one internal standard (ISTD).

Compound	Acronym	Transition Qualifier (*m/z)*	Transition Quantifier (*m/z*)	Retention Time (min)
Targets				
Perfluorooctanoic acid	PFOA	413.00 > 169.05	413.00 > 368.95	8.6
Perfluorooctane sulfonate	PFOS	499.00 > 98.90	499.00 > 80.15	8.9
Perfluorobutane sulfonate	PFBS	299.00 > 99.10	299.00 > 80.10	6.8
ISTD				
Perfluoro-n-[1,2,3,4-^13^C_4_] octanoic acid	M2PFOA	414.80 > 169.00	414.80 > 369.90	8.7

**Table 2 toxics-07-00018-t002:** Concentration of PFOA, PFOS and PFBS in river water (ng/L).

Compounds				Regimes
Sample ID	PFOA	PFBS	PFOS	
RW1	76.79	8.59	0.08	Regime A
RW2	86.69	20.75	ND	
RW3	66.44	6.78	0.12	
RW4	98.21	3.82	ND	
RW5	107.82	3.88	<LOD	
RW6	97.82	2.59	ND	
RW7	105.12	4.26	0.06	
RW8	95.81	1.72	ND	
RW9	1.15	1.24	<LOQ	
RW10	3.65	2.41	0.06	
RW11	1.56	1.89	0.10	Regime B
RW12	<LOQ	2.99	<LOQ	
RW13	<LOQ	3.49	0.06	
RW14	<LOQ	2.12	<LOQ	
RW15	<LOQ	3.44	0.06	
RW16	3.76	5.29	0.06	
RW17	1.20	4.83	<LOQ	
RW18	<LOQ	5.16	<LOQ	
RW19	0.71	4.61	<LOQ	
RW20	4.35	3.77	<LOQ	

RW: river water; ND: not detected; <LOD: below the limit of detection; <LOQ: below the limit of quantification.

**Table 3 toxics-07-00018-t003:** Summary of studied plant samples (*Tagetes erecta* L.), with their PFAS concentrations (ng/g) and bioconcentration factor (BCF).

Average PFAS Conc./*n* = 20/Water (ng/L)	Plant Samples	PFOA/BCF		PFBS/BCF		PFOS/BCF	
PFOA (37.6)	CS1	48.70	1.30	0.75	0.16	0.41	13.67
	CS2	58.96	1.57	1.44	0.31	1.29	43.00
	CS3	94.83	2.52	1.15	0.24	2.17	72.33
PFBS (4.7)	S4	32.36	0.86	1.44	0.31	0.12	4.00
	S5	34.55	0.92	0.25	0.05	3.57	119.00
PFBS (4.7)	S6	37.34	0.99	0.74	0.16	5.03	167.67
	S7	28.49	0.76	0.45	0.10	4.24	141.33
	S8	18.05	0.48	0.51	0.11	1.39	46.33

**Table 4 toxics-07-00018-t004:** Comparison of PFOA, PFOS and PFBS levels (ng/L) in rivers from previous studies.

River	Country	Sampling Year	Level	PFOA	PBFS	PFOS	Reference
Salt	South Africa	2017	mean	107.82	20.75	0.12	This study
Salt	South Africa	2014	mean	390.0	n/a	46.8	[29]
Diep	South Africa	2014	mean	314.4	n/a	181.8	[29]
Eerste	South Africa	2014	mean	145.5	n/a	22.5	[29]
Yangtze	China	2016	mean	13.5	1.84	1.83	[47]
Yellow	China	2016	mean	2.05	0.99	1.84	[47]
Pearl	China	2016	mean	7.45	4.49	11.09	[47]
Kakum	Ghana	NI	mean	167.4	n/a	113	[49]
Tai	China	2012	mean	24.7	3.18	9.78	[50]
Liao	China	2016	mean	8.95	0.94	3.46	[47]
Ganges	India	2014	mean	1.2	n/a	1.7	[50]
Guadalquivir	Spain	NI	mean	11.6	10.1	1.8	[51]
Orge	France	2011	mean	9.4	4.4	17.4	[52]
Rhine	Europe	NI	mean	4.72	21.28	ND	[46]
Swedish	Sweden	2013	mean	4.2	n/a	6.9	[53]
Pearl	China	2013	mean	3.13	ND	2.2	[54]

n/a = not analysed; NI: not indicated; ND = not detected.

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
