# Peer review of "Propensity of Tagetes erecta L., a Medicinal Plant Commonly Used in Diabetes Management, to Accumulate Perfluoroalkyl Substances"

_toxics, 2019, doi:10.3390/toxics7010018_

Round 1

Reviewer 1 Report

The manuscript reports on the bioaccumulation of PFAs (pentafluoroalkylustances) present in water  into Tagetes erecta L. The methology sounds and the method of quantification has been tested for linearity, precision and recovery. The results are interesting and it would be worthy to add a paragraph/reference on the safety level of toxicity of these compounds for humans.

Comment: Pag 5, Line 147: the temperature of the LC column which has been used for the analysis is very high (the recommended max temperature from the vendor is 60°C). Is there a specific reason?

Author Response

Point 1: The manuscript reports on the bioaccumulation of PFAs (pentafluoroalkylustances) present in water into Tagetes erecta L. The methodology sounds and the method of quantification  has been tested for linearity, precision and recovery. The results are  interesting and it would be worthy to add a paragraph/reference on the safety level of toxicity of these compounds for humans.

Response 1: A section has been included in the manuscript addressing the suggestion made by the reviewer (see Page 2, from line 71 to 75)

Point 2: Page 5, Line 147: the temperature of the LC column which has been used for the analysis is very high (the recommended max temperature from the vendor is 60°C). Is there a specific reason?

Response 2: We totally agree with the reviewer as to the 80°C which was reported in the previous version of the manuscript being very high. We hence would like to indicate that, it was a typing error, as 40°C is the temperature was used, and the current version of the manuscript has since been corrected, accordingly (see Page 5, line 150). We thank the reviewer for noticing this error.

Reviewer 2 Report

In the current research, authors have shown the tendency of Tagetes erecta L., a commonly used medicinal plant, to accumulate perfluorooctanoic acid, perfluorooctane sulfonate and perfluorobutane sulfonate determined by LC-MS/MS technique. The authors have also shown that the river water used for irrigation of Tagetes erecta is also contaminated with perfluorooctanoic acid.

Please see the comments below

Line 140-174:

How was the internal standard prepared and why is the injection volume of ISTD (5ul) and injection volume of sample different (10ul)? Since the amounts of detection is in nanograms the effect of contamination is very high. Hence the authors should also run a blank just with MeOH and Milli-Q-water and check the background signal and compare with the samples injected.

What was the concentration of perfluorooctanoic acid, perfluorooctane sulfonate and perfluorobutane sulfonate present in river water?

Author Response

Point 1: How was the internal standard prepared and why is the injection volume of ISTD (5ul) and injection volume of sample different (10ul)? Since the amounts of detection is in nanograms the effect of contamination is very high. Hence the authors should also run a blank just with MeOH and Milli-Q-water and check the      background signal and compare with the samples injected.

Response 2: First of all, we apologise as we seem not to properly comprehend what the reviewer is addressing. However, if we can attempt to answer, we would say that, the internal standard was purchased in pre-prepared solution form, and only known volume(s) was/were used, where necessary. The only explanation we can come up with is that the same injection volume of 10 ul was used for all the sample types analysed throughout the study. As such, while the samples were being prepared, a known volume of the internal standard solution was used to reconstitute the dried sample extracts. As for solvent blanks, in order to have the same concentration of the ISTD as with other samples, the solvent blank was prepared by measuring 195 ul of 10% acetonitrile and 5 ul of 2000 ng/ml of the ISTD to make a final volume of 200 ul.

Point 2: What was the  concentration of perfluorooctanoic acid, perfluorooctane sulfonate and perfluorobutane sulfonate present in river water?

Response 2: The concentrations for these three substances (i.e. PFOA, PFOS and PFBS) can be found in Table 2 (see Page 4, Line 228)